# Effect of Lighting Methods on the Production, Behavior and Meat Quality Parameters of Broiler Chickens

**DOI:** 10.3390/ani14121827

**Published:** 2024-06-19

**Authors:** Tibor István Pap, Rubina Tünde Szabó, Ákos Bodnár, Ferenc Pajor, István Egerszegi, Béla Podmaniczky, Marcell Pacz, Dávid Mezőszentgyörgyi, Mária Kovács-Weber

**Affiliations:** 1Department of Animal Husbandry Technology and Animal Welfare, Institute of Animal Sciences, Hungarian University of Agriculture and Life Sciences, Páter Károly 1, 2100 Gödöllő, Hungary; pap.tibor.istvan@uni-mate.hu (T.I.P.); szabo.rubina.tunde@uni-mate.hu (R.T.S.); bodnar.akos@uni-mate.hu (Á.B.); pajor.ferenc@uni-mate.hu (F.P.); mezoszentgyorgyi.david@uni-mate.hu (D.M.); kovacs-weber.maria@uni-mate.hu (M.K.-W.); 2AgriSearch Hungary Kft, Hősök u. 85, 2119 Pécel, Hungary; bela.podmaniczky@gmail.com; 3Led-Lighting Kft, Röppentyű u. 65-67, 4/401, 1139 Budapest, Hungary; pacz.marcell@gmail.com

**Keywords:** lighting, LED, chicken behavior, animal welfare, production, meat quality

## Abstract

**Simple Summary:**

Light is a very important parameter in poultry farming. It can influence production, meat quality and welfare. The increasingly popular energy-efficient light-emitting diode (LED) lighting has replaced other light sources in many farms. We aimed to investigate the effects of incandescent light (IL) and LED lighting on production, meat quality parameters and behavior. In the first five weeks, there was a significant difference in body weights between the groups. The feed conversion ratio was more favorable in all weeks in the LED group. Among the meat quality parameters, only the shear force values differed significantly between the groups. Broilers raised under LED lighting were more active, spent more time socializing and rested less. The results indicated that LED lighting positively impacts animal welfare and production efficiency.

**Abstract:**

Many farms have been replacing traditional lighting sources with light-emitting diode (LED) bulbs because of technological modernization. We aimed to investigate the effects of incandescent lighting (IL) and LED lighting on Cobb 500 broiler chickens for six weeks. Production parameters (body weight, feed consumption, feed conversion ratio), calculated slaughter values (yield%, relative breast%, thigh%) and breast meat quality parameters (pH at 45 min and 24 h postmortem, color, drip loss, kitchen equipment losses, shear force, meat composition) were recorded. Non-stop recordings were used to analyze the behavior of the birds during several periods of rearing. The LED group was significantly better in the body weight parameter between week 1 and 5 and the feed conversion ratio between week 2 and 3. The most significant difference in behavior was observed in the middle of the rearing period. The chickens in the LED group spent more time eating, drinking and interacting, and rested less. There was no difference in the meat quality parameters; only shear force was significantly lower in the LED group (1781.9 g/s vs. 2098.8 g/s). According to our results, LED lighting can bring about positive changes in animal production efficiency, behavior and other important characteristics for meat consumers.

## 1. Introduction

Light plays one of the most important roles in living organisms. In the case of birds, it is more emphatic [1,2]. Several studies have proven that ideal lighting can greatly impact physiological and production parameters, too [3,4,5,6]. Most experiments with light have focused on its duration and intensity [7,8,9]. These are not the only important parameters during rearing or breeding.

Lighting characteristics, such as wavelength, intensity and duration, significantly impact the development, behavior and welfare of poultry [10,11]. Ma et al. [12] found that laying hens prefer light intensities of 30 lux or less compared to 100 lux. Given a free choice, they spend most of their time at lower illumination levels (5 lux—45.4%; 15 lux—22.2%; 30 lux—22.1%), while they spend less time at higher light intensities (100 lux—10.4%) under fluorescent tubes. Franco et al. [13] confirmed that there are also animal welfare effects of lighting color. They conducted studies with several light wavelengths and found that white light enhanced activity while blue light increased resting state and decreased stress levels in broilers. White monochromatic LED light positively affects chicken meat, improving the amino acid content [14]. Green light has been shown to increase growth hormone-releasing hormone in the hypothalamus and plasma growth hormone concentrations in broilers exposed to LED with different wavelengths. This is because green light activates the secretion of plasma melatonin, which is essential for photoelectric conversion [15]. Bennato et al. [16] determined that the lighting source did not affect carcass yield %, cooking loss %, pH after 24 h, and total lipid parameters. Warm LED light (K = 2500–3000) could significantly increase live weight and decrease carcass weight and L* values compared to neon lighting. Neutral (K = 3500–3700) and cool LED light (K = 5500–6000) significantly increased drip loss % compared to the value (0.90 ± 0.14%) for neon lighting. Colapietro et al. [17] measured similar parameters with control neon and three different LED lights. There were no differences in the case of moisture, total lipids, and dry matter; however, in the neutral LED group, cooking loss was significantly higher among the other experimental and control groups.

Lighting color is based on the wavelength of the light. Light wavelength gives lighting its color. The wavelength of visible light is between shorter invisible ultraviolet (UV) and longer invisible far-infrared rays (FIR), ranging from 380 nm to 740 nm. In contrast, birds can perceive a wider spectrum of light [4,18,19,20,21]. Numerous studies have shown that birds can see in the UVA range [19,22].

Several studies have focused on the effects of different kinds of combined monochromatic light. Combining green and blue monochromatic light can improve broilers’ stress response and immune function [23]. This mixed lighting benefits body weight, muscle growth and meat quality [24]. In green × blue mixed LED light, the final weight of birds increased by 10.66% compared to white light [25]. The quality of broiler meat can also be improved by green light. Green light reduced cooking loss (CL) by 9.9% and increased pH by 1.69% and shear force (SF) by 13.9% compared to white light. [25]. Green and blue light revealed higher pH, water-holding capacity and protein content in the breast compared to red light. However, red light reduced cooking loss, lightness value (L*), shear value and fat content [24].

Lighting also has an effect on behavior. In the case of behavioral studies, the position of selected birds is automatically tracked more frequently. The possibility of monitoring individual animals rather than entire flocks is one of the benefits of tracking technology. To ensure the quality of the research, the transmitters or tags of the automated positioning system affixed to the birds should not impact the behavior, welfare or productivity of the birds [26].

In recent years, there has been much research to examine the effects of lighting on the behavior of poultry. Kristensen et al. [27] examined the free choice of broilers among three lighting sources: one incandescent light and two fluorescent tubes (Biolux and warm white tubes). There was no significant difference at the beginning of rearing, but in the second rearing stage, the birds spent significantly more time under the two fluorescent tubes. The preening was more intense in the Biolux light than in the warm white light. This is possible because the UVA light that Biolux lighting produces has the potential to alter the reflectivity of feathers and the appearance of the testing rooms. Widowski et al. [28] also found that laying hens prefer a compact fluorescent lamp to an incandescent lamp when given a free choice.

Monochromatic green LED light has a positive effect on exploratory behavior. Hens spent more time pecking at objects. Compared to the white light LED, the red light LED decreased aggression [29]. In contrast, Sultana et al. [30] found that the birds in green and blue monochromatic LED light showed more sitting and standing behavior, while in red and red-yellow monochromatic LED light, they showed higher walking behavior.

Our aim was to understand the effects of complex-spectrum LED lighting and incandescent light on broiler chicken growth, feed consumption, meat quality parameters and behavior. Most studies focus on production parameters, meat quality or behavioral parameters. We attempted to comprehensively explore and integrate these aspects within a single study.

## 2. Materials and Methods

### 2.1. Experimental Design

All procedures relating to the use of live birds in this study were carried out with the knowledge and permission of the Institutional Animal Welfare Committee of Hungarian University of Agricultural and Life Sciences Szent István Campus (certification No.: MATE-MKK-2020/22) following the European guidelines for the care and use of animals in research [31].

Cobb 500 male chicks (n = 400) were used in our experiment at the reference farm of the Hungarian University of Agriculture and Life Sciences (Budapest, Pest County, Hungary; GPS coordinates: 47.500972 N, 19.304372 E). The birds were distributed in two groups with five replicates (n = 40) each, as follows: incandescent light (IL) and light-emitting diode (LED).

The incandescent light (IL) group used HELIOS^®^ Soleo light bulb (Katowice, Poland) (60 W, 50 Hz). The LED group used BrightLife^®^ LED (Budapest, Hungary) (9 W, 100,000 Hz). The duration and intensity of lighting were the same in the two groups. The lighting period was decreased for the first five days (Day1 L23:D1; Day2 L22:D2; Day3 L21:D3; Day4 L20:D4; Day5 L19:D5), followed by the L18:D6 photo-period until the 42nd day of life (light:dark hours; L:D). In the LED group, 5 min sunrise and sunset periods were used. The lighting intensity was 60 lux from 1 to 3 d, 40 lux from 4 to 10 d, 30 lx from 11 to 21 d, and 25 lux from 22 to 24 d of age. Both lights could be dimmed to achieve the same intensity of light, and this feature was used to reduce the brightness. Two different luxmeters were used to measure the luminous intensity (Voltcraft^®^, LED luxmeter MS-200LED, and Voltcraft^®^, luxmeter LX-10, Conrad Electronic SE, Hirschau, Germany). The difference between LED and incandescent light was in their wavelengths. The color temperature of the incandescent light was 2700 K. The color temperature of the LED light was 4000 K. The wavelengths of the light employed by the experimental groups are shown in Figure 1. The lighting parameters complied with legal requirements, which also cover the length and intensity of the lighting [32].

The wavelength of light was measured at the same level as the broilers’ heads using the Ocean Optics^®^ USB 2000+VIS-NIR instrument (Ocean Insight, Orlando, FL, USA), which covers a 350–1000 nm wavelength range. Illumination was measured in full light before stocking. The graph does not change with intensity, but the ratio does. At the beginning of rearing, the light intensity is higher, so the intensity values (Figure 1) are higher. In the later stages of rearing, the light intensity decreases, so the intensity observed in Figure 1 also decreases. Despite the change in light intensity, the graph, i.e., the wavelength image, does not change.

In a trial lasting 6 weeks (42 d), 1-day-old Cobb 500 (Cobb-Vantress Inc., Siloam Springs, AR, USA) male chicks were purchased from a commercial hatchery. The male chicks were randomly housed in the two environmentally controlled areas (5 × 3 m^2^ (1.5 m width × 2 m depth)), and the density of the broilers was 14 birds/m^2^. The two groups were housed in the same barn. At the start of the experiment, the house temperature was kept at 33 °C and then reduced as the birds aged. During the last 3 weeks, the temperature was above the recommendation (Table 1). This was especially true during the last week, when the birds were under heat stress and cooling was inadequate. The room temperature was recorded every day using a min/max thermometer (Kerbl, Buchbach, Germany). The barn was separated by a light trap wall in the middle section. Deep litter housing was applied in this trial; each experimental area contained fresh pine shavings at a depth of 8 cm, poultry tube feeders, and a 5-nipple drinking system [33]. The birds were provided a 3-phase feeding program (starter: 1–14 days; grower: 15–28 days; finisher: 29–42 days). Diets were formulated to match the dietary recommendations of the Hungarian Feed Codex [34] (Table 2). Crumbles were used for the starter feed, and whole pellets were used for subsequent feeding. Water and feed were offered for ad libitum consumption. 

Feed consumption of the animals was monitored weekly (day 7, 14, 21, 28, 35, 42). The body weight was measured weekly (day 0, 7, 14, 21, 28, 35, 42). Each animal was weighed, and the average body weight was calculated. The FCR of the birds was calculated from the results of the body weight measurements and weekly feed consumption per group.

### 2.2. Mortality

The mortality is shown in Table 3. The mortality is shown as a percentage of the group and as a number of individuals per group. On day 0, there were 40 birds in each replicate. By day 7, 1 chick died in each group, i.e., an average of 0.5% mortality in each group. In the second week (days 7–14), 5 chicks each died in both groups, giving a mortality of 3% by the second week. In the third week (days 14–21), 1 chick died in the LED group and 3 in the IL group. In the fourth week (days 21–28), 1 chick died in the LED group and 0 in the IL group. In the fifth week (days 28–35), no chicks died in either group. In the sixth week (days 35–42), 6 chicks each died in both groups. At the end of rearing, the total mortality in the LED group was 7%, while in the IL group, it was 7.5%. There was no statistical difference between the number of chicks in the groups for any week. All other calculated values (body weight, feed consumption, etc.) were corrected for mortality on each day.

### 2.3. Examination of Behavioral Characteristics

Behavior was measured by ethological observation, including the percentage of the day spent eating, drinking, resting and interacting.

During the rearing, non-stop video recordings were made one after the other for each group. We analyzed 10 days from the first part of the raising period, 5 from the middle part and 3 from the last part. The first 5–14 days were regarded as the first period, 21–25 days as the middle period and 35–37 days as the last period. The whole pen area is clearly seen in the video.

The video recordings were analyzed using VLC media player software (VLC 3.0.21 Vetinari) according to the following pattern: every 5 min during the illuminated period (18 h), the recording was stopped and the individuals taking part in the following activities were counted: eating time, drinking time, resting time, and entering into interactions (Figure 2); and the number was subtracted from the total number of chickens. In this way, the number of chickens that engaged in activities other than standing, moving and dust-bathing was also calculated.

−Eating time: the activity of standing next to the feeder and putting the head inside.−Drinking time: the activity of standing under the drinker and raising the head to a nipple.−Resting time: the activity when the animal was lying in one place.−Interaction: defined as the activity of one or two birds jumping on each other.

The given time second was recorded in a 10 s time window, so that we could be sure that the activity actually existed (for example, in a still image, it is difficult to determine which animal was moving). The daily results were averaged. The time spent on each activity was given as a percentage of the daily activity. The same person observed the video recordings each time.

### 2.4. Slaughter Procedure

The body weight of the broilers was measured before the slaughter. After weighing the broilers (n = 20/experimental group), they were transported by hand to the experimental slaughterhouse located next to the rearing area. The extermination involved cutting the carotid artery complex along the neck. This complies with the animal experimentation regulation [35]. Their extermination and feathering were followed by repeated weighing (slaughtered body weight), as well as evisceration (whole carcass weight). The carcasses were cooled to a core temperature of 4 °C.

During filleting and cutting, the breast fillet (boneless and skinless) and the leg (with bone and skin) were weighed.

The whole slaughter procedure was conducted according to Hungarian National Standard practices.

### 2.5. Meat Quality Analysis

Meat quality tests were performed on 20 birds from each group, with 4 randomly selected individuals per replication. Both breasts of the chickens were used for the meat quality tests: 10 half right breasts per group were used for pH, color and meat composition parameters, and the remaining 10 half right breasts per group were used for drip loss analysis. The samples were homogenized for meat composition measurement. The other 20 half left breast samples per group were stored at −20 °C before further laboratory testing (thawing loss, cooking loss, cooling loss, shear force).

The yield % was calculated using the formula shown, where the whole carcass represents the slaughtered and gutted body:yield % = whole carcass weight/slaughtered body weight × 100(1)

A HANNA^®^ pH meter (Hanna Instruments Inc., Smithfield, RI, USA) was used to determine the pH values accurately. Prior to measurement, the meter was calibrated in pH 4.01 and pH 7.01 reference solutions for accurate measurement. The measurements were performed on breast meat samples at 45 min after slaughter (pH45) and on chilled bodies (4 °C) at 24 h after slaughter (pH24).

Color was measured on the fresh-cut surface of the breast meat using the CIELAB system. The measurements were performed using a Minolta Chromameter CR 410 (Konica Minolta Inc., Osaka, Japan) colorimeter with a 50 mm head (2° standard observer, C light source) in the CIE L* a* b* color system, where L* is meat color lightness (0 = black; 100 = white), a* is redness (+ red; − green) and b* is yellowness (+ yellow; − blue). Before every measurement, the colorimeter was calibrated against a white calibration plate.

The overall color difference (ΔE*) between the groups was calculated using the formula below [36]:ΔE*_ab_ = ((ΔL*)^2^ + (Δa*)^2^ + (Δb*))^2^1/2(2)

The Lukács [37] visual perceptibility scale was used to assess the total color difference (ΔE*), where ΔE* less than 1.5 means not perceptible, ΔE* from 1.5 to 3 means perceptible, ΔE* from 3 to 6 means well perceptible and ΔE* more than 6 denotes great perceptibility.

After the color measurement, the meat composition values of the breast samples were tested. The meat samples were homogenized with a hand-held blender (HR1600 Pro Mix Daily Collection, 550 W, Philips, Amsterdam, The Netherlands) and placed in a test vessel. After surface homogenization was determined using near-infrared spectroscopy (NIR, Perkin Elmer DA6200, PerkinElmer Inc., Shelton, CT, USA), the chemical composition was noted in percentage, such as moisture, protein, fat, collagen, ash and salt content.

Water-holding capacity was determined using breast meat, with the Honikel test [37]. After cutting, samples of around 100 g each were placed in a 4 °C space on plastic hooks to hang by their own weight, excluding any external influence (except gravity). The mass of the samples was measured every 24 h for three days. Drip loss was expressed as a percentage of weight loss in 72 h compared with the initial sample weight.

After one month of freezing (−20 °C), all frozen samples were thawed at 4 °C for 12 h and at room temperature (22 °C) for 2 h. Then, the meat quality (thawing loss, cooking loss, cooling loss, shear force) was assessed.

Thawing loss was calculated as a percentage of weight loss before and after thawing. Cooking loss was determined as described by the AMSA [38]. The meat samples were baked in a contact grill oven (Cucina HD 2430, Philips, Hamburg, Germany) to a core temperature of 72 °C. In the center of the meat sample, temperature was measured with a digital thermometer (DET1R, Voltcraft, Hirschau, Germany). After the heat treatment, the samples were re-weighed and gently cooled to room temperature (22 °C) for 1.5 h. Cooking loss was calculated as the weight loss percentage before and after heat treatment. Cooling loss was calculated as the weight loss percentage before and after cooling. From the roasted and cooled breasts, 2 test specimens each of 1 × 1 cm pieces were cut. Shear force was measured at 5 points on each piece (n = 200) using a TA.XT PLUS (Stable Micra System Ltd., Godalming, Surrey, UK) texture analyzer equipped with a Warner Bratzler blade. Each specimen was positioned perpendicular to the cutting blade, which passed through the specimen at a speed of 250 mm/min. The texture analyzer was equipped with a 50 kg cell, and the force was calculated based on the force per unit time (g) diagram using Texture Exponent 32 (Stable Micro System Ltd., Godalming, Surrey, UK) software.

### 2.6. Statistical Analysis

The statistical analysis was processed using the R software package R version 4.3.2 (2023-10-31 ucrt) (Core Team R, Vienna, Austria, 2013) [39]. The normal distribution of the groups was checked using the Shapiro–Wilk test, and the F-test was o test the homogeneity of variance. The statistical evaluation between the two groups was evaluated by *t*-test using a two-tailed test. The level of significance was based on *p* < 0.05. Correlation analysis was performed to the meat quality parameters using Pearson’s correlation with pairwise comparisons to determine simple correlation coefficients.

## 3. Results

### 3.1. Production Parameters

As shown in Table 4, in each week of rearing, the birds in the LED group gained a higher body weight. In the first five weeks, the two groups had a significant difference in body weight (*p* < 0.05). The average body weight difference is also shown in Table 4, where the IL group has a higher initial weight (0.3 g). In each additional week, the LED group achieved a higher weight, which was 43 g at the end of week 6.

Feed consumption (Table 5) varied over the weeks. However, there was a significant difference in week 6, when the LED group consumed less feed than the IL group.

The LED group had a significantly better feed conversion ratio on days 8–14 and 15–21. From day 22 onwards, there was no significant difference between the two groups.

The average total feed consumption of the LED group over the 42 days of rearing was 4273.7 g per bird, resulting in a feed conversion rate of 1.67 kg/kg. For the IL group, the average feed consumption was 4335.7 g per bird, resulting in feed sales of 1.72 kg/kg. There was no significant difference between the values of the two groups (N.S.).

### 3.2. Behavior Parameters

The results of the investigated behavioral parameters (average percentage of eating time (a), drinking time (b), resting time (c), other activities (d) and interactions (e) of the birds in the group) are presented in Figure 3a–e.

During the first part of the rearing period (days 5–14), the eating time showed opposite trends between the groups. Drinking time, resting time and other activities also differed during the first part of the rearing period. Interactions between the birds were the same on day 12 of the first period, except for day 6, with the LED group spending more time interacting on all subsequent days, showing a significant difference on days 8 and 13.

In the middle of the rearing period, five days were tested (21–25 d). The eating time spent on all five days was significantly higher for the LED group. The LED group spent significantly more time drinking on days 22 and 25. The resting time was significantly higher in the IL group on all five days. The time spent on other activities showed opposite trends between the groups during the five days examined in the middle phase of the rearing. Bird interactions were higher in the LED group on all five days, showing a significant difference on day 22.

During the last part of the rearing period, three days were examined (days 35–37). The eating time on all examined days was greater for the LED group, with a significant difference on day 36. However, drinking time and resting time show opposite trends between the groups. The time spent on other activities on the days studied was significantly higher in the IL group. No interaction was observed between the birds in either group on days 35 to 37, with a great part of their time spent resting.

### 3.3. Meat Quality Parameters

The meat quality parameters are shown in Table 6. In the case of the meat quality parameters, there were no significant differences between the two experimental groups except for the shear force. The yield %, relative breast % and relative thigh % were more favorable in the LED group. The drip loss % difference between the groups was statistically insignificant.

Thawing loss, cooking loss, cooling loss and the total kitchen losses (including the thawing loss, cooking loss and cooling loss) showed statistically insignificant differences between the groups. Shear force was significantly higher in the IL group (2098.82 g/s). There was no significant difference in any color parameter. The visual perceptibility scale was found to be ΔE* = 0.45, with no difference perceptible to the eye.

The meat composition parameters can be found in Table 7. No significant difference was observed for any of the parameters (moisture, protein, fat, collagen, ash, salt).

### 3.4. Results of Correlations between Meat Quality and Meat Composition Parameters

The correlation analysis values are shown in Table 8. The orange part above the diagonal is IL correlation parameters, and the blue part below the diagonal is LED correlation parameters. All values where negative or positive differences were obtained are indicated, but only the more important correlations are discussed in the results.

## 4. Discussion

There are many ways to improve broiler production efficiency, with varying degrees of success. In our study, a change in housing technology was used instead of different feed ingredients and feed supplements. Our goal was to identify the impact of different light sources (LED and IL) on broiler production parameters, behavior and meat quality.

Improving animal welfare parameters, which are so important for consumers, producers and researchers, can also contribute to more efficient production. An observational study was carried out and compared with production values to support this. During the 6 weeks of rearing, the LED group consistently produced higher body weights. The sudden onset of warm weather can explain the decrease in feed intake in week 6 in both groups. The feed conversion ratio was significantly more favorable on days 8–21 in the LED group. This trend is similar to the results of Mendes et al. [40] with compact fluorescent lamps (CFLs) and LED light sources. The birds in the LED group generally had better production performance than those in the CFL group. Similarly, Archer [41] and Olanrewaju et al. [42] reported that birds reared under LED lighting had higher body weights and better feed conversion than those reared under incandescent lighting. In contrast, Rogers et al. [43] found no significant difference in body weight and feed conversion parameters between the LED and incandescent light groups.

Our observational study showed that the behavioral characteristics of the LED group were favorable because the birds spent less time on other activities and more time on activities that increase production efficiency, such as eating and drinking. This meant that broilers consumed more feed, but also improved the feed conversion due to the background digestive processes, such as rest, which promotes feed utilization and conversion [44]. This connection demonstrates that the feed efficiency can be increased by changing the housing parameters due to the habits and welfare of the birds.

Of the days studied, significantly more interactions were observed in the LED group on days 8, 13 and 22. This is assumed to represent a more active exercise of species-specific behavioral traits, rather than the birds becoming more stressed [29]. Archer [41] conducted fear and stress susceptibility tests investigating the effects of incandescent light and two different LEDs. It was clearly demonstrated (*p* < 0.05) that birds in the two LED groups showed lower stress susceptibility and fear. In contrast, Olanrewaju et al. [42] found no significant difference in welfare indicators (eyes to BW, humoral immune response, ocular assessment, ocular histopathologic examination, etc.) between the two LED groups and the incandescent light group tested. Rogers et al. [44] and Archer [41], examining LED and compact fluorescent lamps, concluded that LED lighting does not increase, and in many cases decreases, stress levels in birds. These results also suggest that modern LED technology does not adversely affect bird welfare.

The changes in the middle and final stages of rearing seem to be significant for the time spent interacting, eating, resting and on other activities, but this is perfectly normal. The difference between the middle and final stages of rearing is only 1.5 weeks, but the birds double their weight during this period. This means that they gain about 2 kg of live weight instead of about 1 kg of body weight. In addition, if the density of stocking was used in accordance with the rules, the area available for movement of the animals was significantly reduced by the final stage of rearing.

Whenever there is an increase in body weight due to changes in housing and feeding technology, the question always arises whether the visceral tract, carcass or even the amount of valuable meat parts has increased or decreased. The average difference at the end of rearing of 43 g body weight in the present study does not differ significantly. However, for a broiler flock with an average weight of 2500 g, calculated with the weight of the two study groups for 100,000 birds, this represents a body weight surplus of 4.3 t, which could be used in the food chain during one rotation of a given flock. The growth rate of poultry and its breast yield usually do not leave the histological condition of the muscle intact, and the physiological background processes are also subject to change [45]. The rheological properties of the breast meat of broilers, which is considered to be tender, are influenced by the age of the animal and thus, in a variable way, by the moisture content of the product [46].

In many cases, the conditions during the meat maturation process, the amount of lactic acid accumulated and the timing of its production are responsible for the change in protein condition, which in many cases also determines the shear force value. However, our study found no difference in pH values, but a significant difference in shear force values was detected (*p* < 0.001).

The tenderness of meat is important for both the processing industry and consumers, as the quality of processed products is determined by the water-holding capacity of meat proteins during both storage and possible heat treatment [47]. If a sudden decrease in pH is observed at the beginning of the postmortem period (pH45), this will result in shrinkage of myofibrils and affect protein function, leading to a decrease in the water-holding capacity of proteins [48]. There is a strong correlation with meat tenderness due to water loss during processing or cooking [49]. According to Vaskoska et al. [50], although the effect of thermal denaturation on the structure and quality of different types of muscle fibers is not fully understood, it is known that denaturation dynamics can be used to infer microstructural changes and cooking losses, which may be influenced by the type and characteristics of the muscle fiber and by pH alone or in combination with the previous one.

In our study, the statistical differences between the two groups and the correlation values between the parameters in each group also show the relationships discussed in the previous paragraph. In the LED group, a strong negative correlation was observed between collagen and cooling loss (r = −0.95), while a strong positive correlation was observed between collagen and shear force (r = 0.92). Concerning collagen, two correlations were discovered that could suggest further research opportunities: in the LED group, the relative breast (r = 0.99) and thigh (r = 0.91) values showed a strong positive correlation with the collagen content of the breast. These correlations could not be detected in the IL group; however, based on the correlation analysis, the effect of pH45 prevailed much more: a close negative correlation was observed with the ash content of the breast meat of the IL group (r = −0.95), a moderate negative correlation with the lightness of the meat (L*) (r = −0.45) and a medium-strength correlation concerning thawing loss (r = 0.64).

While there was no statistically verifiable difference in pH values, time of pH reduction and kitchen losses between the LED and IL groups, the LED group performed better in tenderness (*p* < 0.001). Significantly lower Warner–Bratzler shear force values were detected (*p* < 0.001). Our results are fully supported by the experiment of Kim et al. [14], where the shear force values in the LED group were significantly (*p* < 0.001) lower than in the incandescent light group, the cooking loss was higher in the incandescent light group, and there was no difference in water-holding capacity.

Changes in the sensory properties of poultry meat are particularly important. In addition to tenderness, the color of meat products is also a very important element, a primary component of their appearance and a significant influence on purchasing behavior. Defects in meat quality can have a major impact on color, in addition to tenderness/softness and shear force values, which are associated with selection work based on broilers’ growth rate and muscle development [51].

No significant difference was found in color between the two groups (L*, a*, b*), and the visual perceptibility scale also proves (ΔE* = 0.45) that the difference is not perceptible for consumers, according to the visual perceptibility scale of Lukács [36]. Ke et al. [52] investigated different (white, red, green, blue) monochromatic LED lights. Compared to our study, a darker breast was obtained using monochromatic light, with L* values ranging from 50.09 to 53.32, while in our study, they were 60.11 and 59.8. In our case, the light PSE-like meat can be explained by the warm weather in the week before slaughter, which kept our birds under heat stress. Kannan et al. [53] concluded that higher plasma corticosterone levels were associated with lighter PSE in breast and thigh meat. In breast meat from heat-stressed turkeys, a greater rate of postmortem pH reduction was detected, resulting in paler meat than in non-stressed turkeys [54]. Thus, seasonal heat stress may play a role in the development of lighter meat by accelerating postmortem metabolism and biochemical processes in muscle.

## 5. Conclusions

In conclusion, LED lighting, which is becoming increasingly popular due to its energy efficiency, can positively change animal production efficiency, behavior and other important characteristics for meat consumers. LED lighting corresponds to high-quality production with favorable animal welfare parameters and is more sustainable from an economic and environmental point of view. Interactions between birds were higher in the LED group, meaning that this species-specific behavior was practiced more. This suggests that the birds felt more comfortable, i.e., animal welfare was improved. It is recommended that the poultry sector pay attention to other lighting parameters (e.g., spectral composition, frequency), and not only lighting duration and intensity, to enhance welfare and possibly improve production indicators. Adequate lighting for birds can improve the feed conversion ratio without negatively affecting meat quality characteristics. Feed conversion was significantly better in the LED group in weeks 2 and 3. There was no difference in mortality between the groups. In the LED group, the kitchen losses were more favorable and the shear force was also lower.

Using LED lighting in broiler production does not decrease the efficiency compared to conventional lighting, but has a positive effect on several parameters. For further experiments, it is recommended to try LEDs with different spectra (including UV and far-red), testing similar production and meat quality parameters.

## Figures and Tables

**Figure 1 animals-14-01827-f001:**
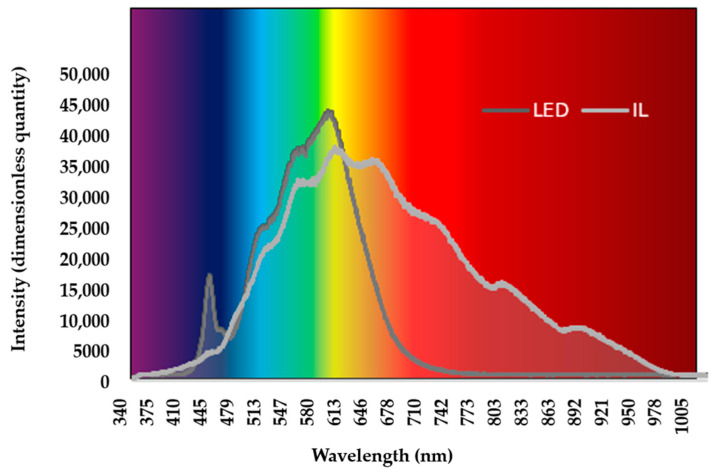
LED and IL wavelengths of average intensity. LED—light-emitting diode; IL—incandescent light.

**Figure 2 animals-14-01827-f002:**
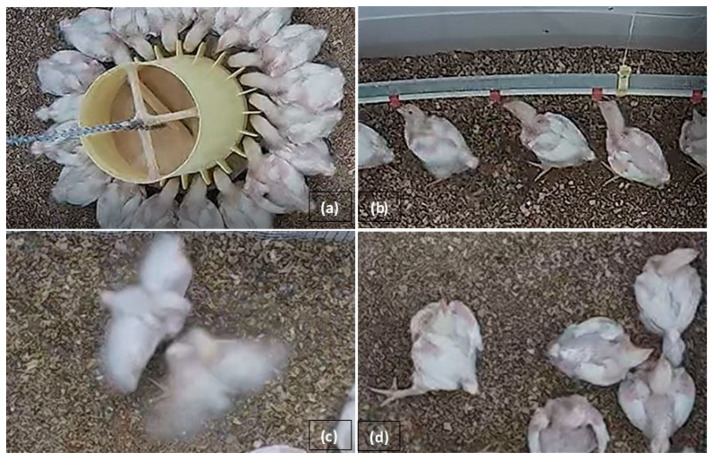
Observed behaviors: (**a**) eating time, (**b**) drinking time, (**c**) interactions and (**d**) resting time.

**Figure 3 animals-14-01827-f003:**
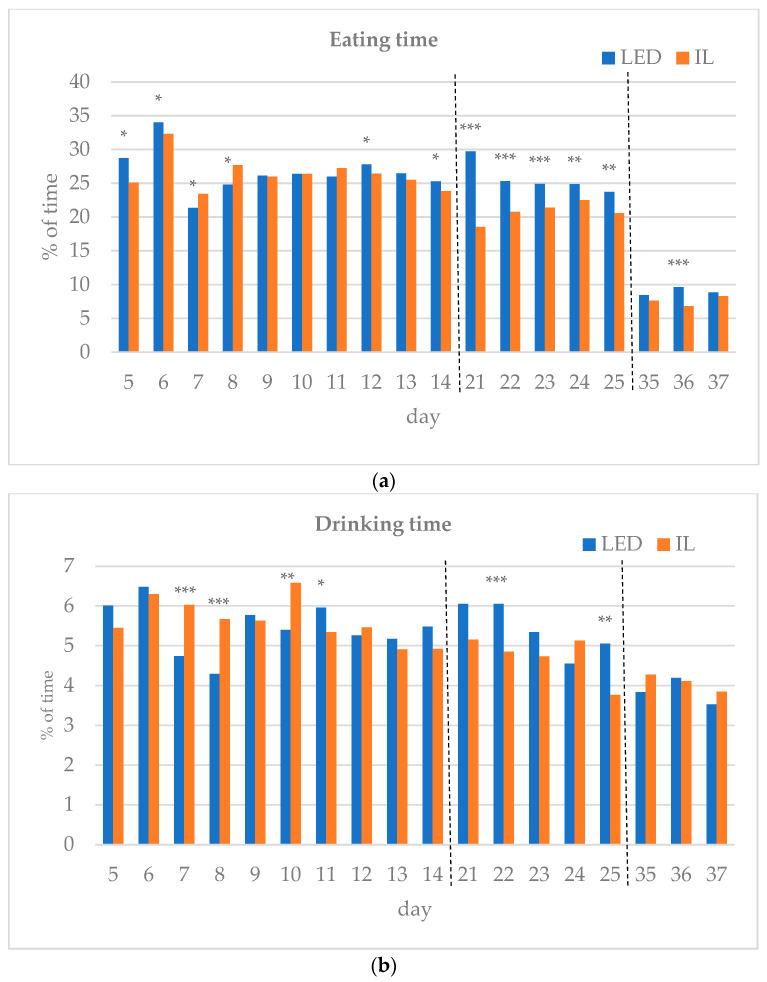
(**a**–**e**) Behavioral parameters of the birds during the rearing period (first part: days 5 to 14; middle part: days 21 to 25; last part: days 35 to 37; part boundaries of investigations are indicated by dashed lines). LED—light-emitting diode; IL—incandescent light. * Figure 3a–e with statistical values (*p*-value: <0.05 *; <0.01 **; <0.001 ***).

**Table 1 animals-14-01827-t001:** Recommendation and real temperature during rearing.

Age—Days	Temperature °C Recommended [35]	Temperature °C Real
0	33	33
7	30	30
14	27	27
21	24	24
28	21	22
35	19	22
42	18	26

**Table 2 animals-14-01827-t002:** Composition of the experimental diets.

Dietary Composition			
	Starter (1–14 d)	Grower (15–28 d)	Finisher (29–42 d)
	Ingredient (%)		
Corn	34.00	38.00	41.00
Wheat	19.00	16.00	18.00
Extracted soy (46%)	31.00	23.00	16.00
Extracted sunflower (37%) (unhulled)	4.00	10.00	11.00
Corn gluten (60%)	4.00	4.00	5.00
Sunflower oil	3.50	5.00	5.30
Premix *	0.40	0.40	0.40
Limestone	1.20	1.10	0.90
L-lysine	0.50	0.40	0.40
DL-Methionine	0.30	0.25	0.20
L-Threonine	0.15	-	-
MCP	1.70	1.60	1.55
NaCl	0.25	0.25	0.25
∑	100.00	100.00	100.00
	Nutrient %		
Dry matter	88.301	88.344	88.166
Metabolizable energy (AMEn) MJ/kg	12.585	13.095	13.412
Crude protein	22.805	21.200	19.455
Crude fat	6.679	8.493	8.721
Crude fiber	3.784	4.635	4.631
Lysine	1.453	1.275	1.126
Av lysine	1.399	1.217	1.120
D lysine	1.255	1.110	0.999
Methionine	0.712	0.634	0.611
Methionine + Cysteine	1.023	0.961	0.873
D-Methionine + Cysteine	0.963	0.885	0.811
Threonine	1.028	0.824	0.756
D-Threonine	0.846	0.756	0.685
Tryptophan	0.286	0.265	0.233
D-Tryptophan	0.215	0.182	0.155
Arginine	1.312	1.243	1.062
Valine	1.078	0.998	0.876
Calcium	0.959	0.885	0.802
Phosphorus	0.889	0.798	0.761
Coccidiostats	+	+	-

* Premix for the starter and grower phases containing (per kg): methionine 3%, calcium 24.30%, phosphorus 5.10%, sodium 4.2%, vitamin A 333,333 IU, vitamin D_3_ 133,350 IU, vitamin E 1125 mg, vitamin K 75 mg, manganese 4000 mg, zinc 3332 mg, iron 1333 mg, copper 600 mg, selenium 10 mg, and phytase activity 16,700 FTU. Premix for the finisher phase containing (per kg): methionine 2.3%, calcium 23.40%, phosphorus 4.55%, sodium 4.1%, vitamin A 200,000 IU, vitamin D_3_ 100,000 IU, vitamin E 1200 mg, vitamin K 60 mg, manganese 3334 mg, zinc 2781 mg, iron 1111 mg, copper 500 mg, selenium 8.33 mg, and phytase activity 16,700 FTU.

**Table 3 animals-14-01827-t003:** Mortality during the investigation.

	Day 1	Day 7	Day 14	Day 21	Day 28	Day 35	Day 42
		Mortality in the Group % (Piece)					
LED	0 (0)	0.5 (1)	3 (5)	3.5 (1)	4 (1)	4 (0)	7 (6)
IL	0 (0)	0.5 (1)	3 (5)	4.5 (3)	4.5 (0)	4.5 (0)	7.5 (6)
SEM		±0.022	±0.043	±0.062	±0.067	±0.067	±0.068
*p*-value		N.S.	N.S.	N.S.	N.S.	N.S.	N.S.

**Table 4 animals-14-01827-t004:** Results of body weight of male broiler chickens by lighting method.

	Day 1	Day 7	Day 14	Day 21	Day 28	Day 35	Day 42
LED	40.4	149.9	422.4	848.0	1403.0	2085.6	2560.0
IL	40.7	142.9	399.3	823.5	1362.5	1978.7	2517.0
^+^LED-IL	0.3	7.0	23.1	24.5	40.5	106.9	43.0
SEM	0.11	0.97	3.01	5.19	8.20	11.50	14.35
*p*-value	N.S.	<0.001	<0.001	<0.05	<0.05	<0.001	N.S.

LED—light-emitting diode; IL—incandescent light; ^+^LED-IL—average body weight difference between the two groups (g).

**Table 5 animals-14-01827-t005:** Results of average weight gain, feed consumption and feed conversion ratio of male broiler chickens by lighting method.

	Days 1–7	Days 8–14	Days 15–21	Days 22–28	Days 29–35	Days 36–42
	Average daily weight gain (g)					
LED	21.41	30.17	40.38	50.11	59.59	60.95
IL	20.41	28.52	39.21	48.66	56.53	59.93
SEM	0.44	1.35	2.34	3.71	5.21	6.48
*p*-value	<0.001	<0.001	<0.05	<0.05	<0.001	N.S.
	Feed consumption (g/bird)					
LED	123.7	335.1	626.9	975.2	1292.4	920.4
IL	121.2	326.4	641.4	965.6	1193.2	1087.9
SEM	0.10	0.42	1.19	1.75	3.50	3.37
*p*-value	N.S.	N.S.	N.S.	N.S.	N.S.	<0.01
	Average daily feed consumption (g/bird)					
LED	17.7	47.9	89.6	139.3	184.6	131.5
IL	17.3	46.6	91.6	137.9	170.5	155.4
SEM	0.05	0.21	0.53	0.76	1.89	2.44
*p*-value	N.S.	N.S.	N.S.	N.S.	N.S.	<0.01
	Feed conversion ratio (kg/kg)					
LED	0.83	1.09	1.29	1.48	1.61	1.72
IL	0.85	1.13	1.33	1.52	1.65	1.77
SEM	0.001	0.001	0.001	0.002	0.001	0.002
*p*-value	N.S.	<0.05	<0.05	N.S.	N.S.	N.S.

LED—light-emitting diode; IL—incandescent light; SEM—standard error mean, N.S.—not significant.

**Table 6 animals-14-01827-t006:** Results of meat quality parameters between the groups.

		LED	IL	SEM	*p*-Value
Yield %		76.41	72.72	1.38	N.S.
Relative breast %		24.46	23.76	0.58	N.S.
Relative thigh %		24.09	23.19	0.43	N.S.
pH45		6.49	6.51	0.03	N.S.
pH24		5.88	5.89	0.04	N.S.
Drip loss %		3.49	2.79	0.28	N.S.
Thawing loss %		3.05	3.41	0.35	N.S.
Cooking loss %		25.27	26.89	0.83	N.S.
Cooling loss %		8.96	9.03	0.20	N.S.
Total kitchen losses %		37.27	39.32	0.91	N.S.
Shear force (g/s)		1781.95	2098.82	0.08	<0.001
Color	L*	60.12	59.80	0.49	N.S.
	a*	12.16	12.28	0.33	N.S.
	b*	11.86	11.56	0.29	N.S.

LED—light-emitting diode; IL—incandescent light; SEM—standard error mean, N.S.—not significant.

**Table 7 animals-14-01827-t007:** Results of meat composition parameters between the groups.

%	LED	IL	SEM	*p*-Value
Moisture	74.42	74.19	0.15	N.S.
Protein	21.42	21.36	0.14	N.S.
Fat	3.20	3.35	0.05	N.S.
Collagen	1.18	1.20	0.01	N.S.
Ash	2.38	2.40	0.02	N.S.
Salt	1.23	1.27	0.02	N.S.

LED—light-emitting diode; IL—incandescent light; SEM—standard error mean; N.S.—not significant.

**Table 8 animals-14-01827-t008:** Correlation coefficients between meat quality and composition parameters (LED group: below diagonal, indicated by blue color; IL group: above diagonal, indicated by orange color).

Item	Yield	R. Thigh	R. Breast	pH45	pH24	Drip Loss	Tha. Loss	Cook. Loss	Cool. Loss	T. kit. Loss.	S. Force	C. L*	C. a*	C. b*	Moisture	Protein	Fat	Collagen	Ash	Salt
**Yield**	-	0.87 ***	0.91 ***	0.10	−0.05	0.30	0.31	−0.45	0.17	−0.17	−0.07	0.24	−0.11	−0.02	−0.51	0.19	0.74	0.21	−0.60	−0.58
**R. thigh**	0.90 ***	-	0.82 ***	0.12	−0.15	0.32	0.31	−0.41	0.09	−0.16	−0.11	0.08	−0.10	−0.09	−0.51	0.18	0.73	0.16	−0.63	−0.58
**R. breast**	0.93 ***	0.69 ***	-	0.15	0.05	0.16	0.17	−0.59	0.16	−0.32	−0.07	0.31	−0.24	−0.09	−0.50	0.18	0.73	0.28	−0.54	−0.58
**pH45**	−0.06	−0.13	−0.01	-	0.16	0.65	0.63 *	0.07	0.24	0.31	0.07	−0.45 *	0.40	−0.07	0.06	−0.36	0.63	−0.68	−0.95 *	−0.70
**pH24**	0.15	0.07	0.18	−0.03	-	0.42	0.13	0.41	−0.10	0.32	−0.15	−0.01	−0.00	−0.12	−0.06	−0.27	−0.54	−0.28	0.18	0.57
**Drip loss**	−0.14	−0.26	−0.00	−0.03	0.57	-	0.62	0.34	−0.12	0.45	−0.37	−0.54	0.26	−0.08	0.38	−0.58	0.58	−0.83	−0.91 *	−0.68
**Tha. loss**	0.26	0.23	0.34	0.11	0.11	0.67 *	-	0.61	0.37	0.83 **	−0.53	−0.67 *	0.67 *	−0.12	−0.17	−0.07	0.34	−0.63	−0.78	−0.36
**Cook. loss**	−0.15	−0.21	−0.16	−0.48	−0.09	−0.25	−0.49	-	0.19	0.94 *	−0.43	−0.70 *	0.42	0.10	0.16	0.05	−0.46	−0.42	0.09	0.45
**Cool. loss**	−0.43	−0.32	−0.48	0.48	−0.09	−0.04	−0.14	−0.40	-	0.40	−0.07	−0.12	−0.01	0.31	−0.46	0.36	0.39	0.03	−0.46	−0.15
**T. kit. loss**	−0.23	−0.28	−0.23	−0.40	−0.11	−0.05	−0.25	0.93 ***	−0.22	-	−0.50	−0.74 *	0.53	0.07	−0.02	0.06	−0.13	−0.55	−0.32	0.14
**S. force**	0.45 *	0.45 *	0.37	0.09	0.33	−0.23	−0.09	0.12	0.09	0.14	-	0.12	0.03	−0.13	−0.17	0.38	−0.24	0.14	0.20	0.44
**C. L***	0.31	0.19	0.36	0.10	0.05	−0.60	−0.45	−0.36	0.18	−0.59	0.13	-	−0.67 **	0.21	−0.23	0.27	0.26	0.86	0.33	−0.04
**C. a***	−0.14	−0.25	−0.24	−0.20	−0.41	0.53	0.25	0.43	−0.14	0.61	−0.37	−0.65 **	-	−0.49 *	0.13	−0.25	−0.21	−0.79	−0.35	−0.04
**C. b***	−0.19	−0.28	−0.07	0.18	−0.23	−0.47	0.01	−0.64 *	0.31	−0.71 *	−0.11	0.62 **	−0.57 **	-	0.28	−0.25	0.59	−0.31	−0.15	−0.42
**Moisture**	−0.76	−0.62	−0.80	0.18	−0.48	−0.30	−0.67	−0.60	0.95	−0.45	−0.60	0.45	−0.21	0.32	-	−0.93 *	0.16	−0.60	−0.20	−0.40
**Protein**	0.56	0.45	0.59	−0.40	0.73	−0.49	0.56	0.63	0.95 *	0.51	0.42	−0.54	0.18	−0.48	−0.91 *	-	−0.44	0.70	0.50	0.68
**Fat**	0.63	0.57	0.70	0.26	−0.30	−0.29	0.40	0.27	−0.64	0.15	0.61	0.04	0.13	0.07	−0.39	0.00	-	−0.14	−0.82	−0.94
**Collagen**	0.97 **	0.91 *	0.99 ***	−0.07	0.37	−0.09	0.36	0.66	−0.95 *	0.33	0.92 *	−0.08	−0.14	−0.25	−0.84	0.66	0.65	-	0.61	0.40
**Ash**	0.18	0.06	0.26	−0.66	0.81	−0.80	0.63	0.65	−0.49	0.70	0.04	−0.80	0.45	−0.69	−0.72	0.91 *	−0.23	0.35	-	0.87
**Salt**	0.27	0.13	0.34	−0.53	0.71	0.73	0.69	0.58	−0.59	0.62	0.10	−0.76	0.45	−0.57	−0.81	0.94 *	−0.13	0.42	0.98 **	-

* **R. thigh**: Relative thigh; **R. breast**: Relative breast; **Tha. loss**: Thawing loss; **Cook. loss**: Cooking loss; **Cool. loss**: Cooling loss; **T. kit. loss**: Total kitchen losses; **S. force**: Shear force; **C. L***: Color L*; **C. a***: Color a*; **C. b***: Color b*; LED—light-emitting diode; IL—incandescent light; * = *p* < 0.05; ** = *p* < 0.01; *** = *p* < 0.001.

## Data Availability

The data presented in this study are available on request from the corresponding author. The data are public.

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
