# Peer review of "Effect of Lighting Methods on the Production, Behavior and Meat Quality Parameters of Broiler Chickens"

_animals, 2024, doi:10.3390/ani14121827_

Round 1

Reviewer 1 Report (Previous Reviewer 2)

Comments and Suggestions for Authors

Dear Editorial Board of Animals

Considering that all my comments have been taken into account and the text have been corrected, I accept the manuscript in its current form.

Kind regards

Author Response

Reviewer 2 Report (Previous Reviewer 1)

Comments and Suggestions for Authors

Dear Author,

The hypothesis of the study is not clearly stated. Although similar studies have been conducted in poultry, the difference of this study from other studies should be stated even in one sentence.

Why did you apply sunrise and sunset in the Led group? Doesn't this application act as a different factor?

Please make sure that the periods in Table 2 correspond to what is written in the text, “starter: 1-14 days; grower: 15-28 days; finisher: 29-42 days”.

In measuring behavioral traits, shouldn't a few animals be marked and only these animals should be monitored for a certain period of time? If you have a reference for this method, please cite it.

Line 229: Delete this sentence “A sharp knife was used to cut the neck”.

Lines 315-317: Please check this sentence. What do you mean here?

Table 5: Why did you not give the body weight gain during the trial period (1-42 days)? It would be more appropriate to give body weight gain, feed intake and feed efficiency during the whole period (1-42 days).

Lines 334-335: “On 7 of the 10 days studied, birds in the LED group spent more eating time.” This sentence is incorrectly written according to the data in the table.

Lines 333-340: Please check these sentences.

Lines 364: Please check these sentences “more drinking time on four days except day 24,”

Table 8: You have to write the results of significant characteristics.

Lines 421-423: This statement contradicts the data in the table. “During the 421 6 weeks of rearing, the LED group consistently produced higher body weights with lower 422 feed consumption.”

Lines 424-426: While only two weeks of feed efficiency are important, it is not correct to attribute it to all periods.

Lines 440-441:  Please add the reference to this sentence. “.This is assumed to represent a more active exercise of species-specific behavioral traits, not that the birds became more stressed.”

Line 455: replace “live” with “body”

Line 463: replace “live” with “body”

Lines 500-501: According to table 6, this expression is incorrect. “the LED group performed better in terms of consumer preferences for overall kitchen losses and tenderness (p < 0.001),”

Line 503: Academically, this statement is incorrect. Statistical differences are important or not It is more accurate to evaluate the results statistically rather than numerically.

Lines 541-545: . This statement is meaningless when you consider the study as a whole. You need to analyze the study separately for 42 days. “Although the results show that feed con- 541 sumption varied over the weeks, there was a significant difference on week 6, where the 542 LED group consumed less feed than the IL group. In addition, body weight and feed con- 543 version ratio were more favorable in the LED group each week, with significant differ- "544 ences in several cases”

Author Response

Reviewer 3 Report (Previous Reviewer 3)

Comments and Suggestions for Authors

I think that revised paper should be accepted for publishing.

Round 2

Reviewer 2 Report (Previous Reviewer 1)

Comments and Suggestions for Authors

Dear Authors,

The corrections made are sufficient. 

Best regards,

This manuscript is a resubmission of an earlier submission. The following is a list of the peer review reports and author responses from that submission.

Round 1

Reviewer 1 Report

Comments and Suggestions for Authors

Dear Authors,

Introduction

This section is thought to be too long and contains sentences that are irrelevant to your study.

Lines 45-60: I think this part is not related to your study. Therefore, it would be better to remove this part.

Lines 99-101: What is the connection between this and your study?

Materials and Methods

Line130: replace “cockerels” with “male chicks”

Lines 137-139: Include the references at the end of the sentence.“The lighting period (light:dark hours; L:D) was”

Line 178: It has been stated that body weight is measured individually. But you didn't specify how you numbered the animals? Why did you do individual weighing?

Didn't you get any data on viability?

Results

Line 287: replace “Table 1” with “Table 2”

Table 2: Initial body weight values of the animals must be given.

Tables: Giving standard deviations in the tables in this way weakens the understandability of the results. It would be more appropriate to give it as standard error mean.

Table 2: It would be more suitable to include the difference between the initial and final body weights in the table.

Table 3: Why didn't you give the standard deviation or error for the behavior?

Table 3: The data in the table regarding behavior are very confusing and poorly understandable. It may be more appropriate to consider these behavioral characteristics as a whole rather than as a day.

Table 4: Please check the color L, a, and b values of the LED groups.

Line 338: “The drip loss % was higher for the LED group.” It is inappropriate to express this sentence. The difference between groups is statistically insignificant.

Lines 339-340: “Thawing, roasting or cooking and cooling losses were higher in the IL group. Thus, 339 total cooking losses were also higher in the IL group (39.32%) compared to the LED group 340 (37.27%).” It is inappropriate to express this sentence. The difference between groups is statistically insignificant.

Lines 341-342: “. In L* and b*, 341 the LED group obtained higher values, while the a* value was lower.” It is inappropriate to express this sentence. The difference between groups is statistically insignificant.

Line 350: Table 5 should display the meat composition. Please verify.

Lines 350-353: It is inappropriate to express this sentence. The difference between groups is statistically insignificant.

In the statistics section, you stated that you calculated correlations between meat quality characteristics. However, the results are not given in the results section.

Discussion

Line 383: please check ttis snetence. There was a significant effects on interaction behavior in just 3 days. On other days, the difference between groups is statistically insignificant.

Line 368: “Feed conversion ratio was more favorable in all weeks”. My opinion is that this statement is misleading. After week 3, the difference between groups is statistically insignificant. Additionally, data should be given between weeks 1-6, that is, during the entire growth period.

 Lines 395-396: There is misunderstanding in this sentence.

Lines 433-444: Correlation data could not be evaluated because they were not given in the table.

Conclusion:

Lines 488-489: Additionally, it is important to highlight other aspects of performance.

Reviewer 2 Report

Comments and Suggestions for Authors

General comment

As the authors of the manuscript state there are many ways to improve the broiler production efficiency, between as different feed ingredients and feed supplements, with varying degrees of success. This study identifies the impact of different light source (LED and IL) to broiler production parameters, behavior, and meat quality, which is a change in the housing technology. The conclusion of this manuscript is that LED light improve interactions between birds, meaning this species-specific behavior was practiced more. This suggests that the birds felt more comfortable, i. e. animal welfare was improved. For this reason, I estimate this work as a valuable.  The manuscript is well written.

Specific comments

1. The aim of the study (lines 119-120): the last sentence is unnecessary. They should be moved to the Material and methods chapter, especially since in the case of the remaining considered parameters, the authors do not provide how they were measured.

2. Material and methods (lines 140-145): it is worth justifying why such light intensities, e. g. 60 lx and color temperatures were used. Especially since they were different for LED and IL. Was this due to the reference/recommended or obligatory/legal values for broiler chickens (if so, the source should be cited) or were there any other assumptions made by the authors (if so, what were they?).

3. Figure 3 (Lines 151-152): does intensity (dimensionless quantity) 0-5000 have no units/abbreviation? It is worth explaining in more detail what this term means and how this parameter is measured. For me (a specialist in animal research), the sentence: 'The graph does not change with intensity, but the ratio does.' is difficult to understand. Can it be explained more clearly/simpler?

4. Cobb 500 (e.g., line 155) is written with or without a space (e.g., line 26).

5. How were the environmental parameters controlled (lines 155-157)? Which devices were used, are they certified? It is difficult to maintain identical environmental conditions without the use of special technical solutions, have such solutions been used, and if so, what ones?

6. Line 210: please provide more details about the method of extermination.

7. The page numbering is incorrect. Either it is missing, or it is repeated.

8. Sentence: 'From the point of view of meat processing technology, it would be worthwhile to consider the influence of different production systems (housing technology + feeding technology) on the shear force value of meat and to select the target processing processes on this basis.' is too general, the housing technology is not only about lighting. It is it worth giving a specific highlight here: how to approach this issue in future research?

Reviewer 3 Report

Comments and Suggestions for Authors

Generally, the paper is well written, and the content of the paper is interesting. Effect of LED light has been well studied so far, but the authors bring a more recent knowledge in a very comprehensive study. This approach gives us a very good insight into the meat production and meat quality of the broilers reared under different environmental conditions.

 I would recommend the paper for publishing with minor revision.

 Specific comments:

Line15: I would suggest saying “…very important parameters…” rather than “… one of the most critical parameters…” There are other parameters in broiler rearing which could be marked as critical, but lighting is not one of them.

Line 35: It is not necessary to put +- SD values in Abstract. Also, there is no need for two decimal places if the value is almost 2000.

Lines 59-61: I suggest merging the last two sentences in one.

Line 106: What is feather-directed? Please use precise terms or explain.

Table 4: The difference in carcass yield is very big (almost 4%) but it is still not significant. We can see that SD is very high which is not typical for this trait. Could you please explain this very high variability inside the groups (in the discussion)?

Lines 398-401: I didn’t understand this part. Please present your statements more clearly.

Lines 406-407: Statement is not clear enough. Please rephrase.

Line 487: Not clear enough. Please do not start the sentence with because.

Reviewer 4 Report

Comments and Suggestions for Authors

Line 140 how did you verify that this intensity was achieved? To achieve this were the lights dimmable or were the total number of lights on varied to achieve the light curve described.

Utilizing various terms to describe the chicks, use one consistently throughout the manuscript.

The description of the behavior observations creates more questions around how the authors determined duration of an activity when they first state they stopped all video every 5 minutes for a count. This section needs heavily revamped. The ethogram is also unclear. if the eating bouts are short for broilers, does this 5 minute interval make sense?

lines 210 to 212 very unclear what data was collected, strongly suggest revising the terminology to improve the perception of the process.

When the authors mention specific brands of equipment the information provided is inconsistent. Please revamp this throughout and make all of them complete

expand on the T-test, 2 tailed or single tailed?

Table 2 make sig figs all the same.

In the discussion the authors should consider the impact of their methodology on the conclusion they draw 

Comments on the Quality of English Language

Multiple phrases throughout, too many to individually note where verb tense is incorrect.

Reviewer 5 Report

Comments and Suggestions for Authors

The authors have tested two different light systems in growing broilers. This is a clear approach with practical relevance. The experimental design is simple, and the methods are well described. The experiments obviously went well. Data on bird mortality, diseases or deformations of animals and other experimental problems are missing or not openly presented (climate) that might explain some variation of results.

Feed intake is probably one of the first factors that influence the performance of the birds. Light as a parameter in this experiment is influencing feed intake directly. This was not sufficiently respected and presented in the paper (Tab. 2).

In Introduction and Discussion contain many aspects and arguments that are not relevant in this experimental design or were not significant in the experiments:
- Laying hens
- meat composition
- Most meat quality parameters
- others

Comments in detail

L. 55               Had higher body weights than ???

L. 62 ff           Laying hens must be discussed separately, whenever!

L. 66 ff           Brightness and colors must be discussed in relation to your experimental treatments.

L. 70 ff           What are the mechanisms between meat quality and light regime.

Tab. 1             * missing in the table: Premix list not clear, what is % and what is … per kg?

L. 285 ff         Add a chapter on mortality, diseases etc.; medical treatments.

Tab. 2             Use Day 7, 14, … instead of Week 1, 2, etc.
Give also data on daily body weight gain
FI: Give data per day in the respective week
FCR: Give data in the respective week.

Tab. 3             A graphic could be more illustrative.

L. 338            Effect was not significant!

L. 350 ff         All effects not significant!

L. 363 ff         Use new data in Tab. 2 for discussion.

L. 367            Warm weather: what does that mean? Were there more side effects that have to be discussed?

L. 379 ff         Your data do not show this effect clearly (side effects?)

L. 398 ff         This section is not clear: what changes?

L. 404 ff         This is a dangerous speculation; one % too much.

L. 421 ff         Effects were not significant.

L. 433 ff         Not clear.

L. 483 ff           This part belongs to the discussion and not conclusions.

Comments on the Quality of English Language

In general OK

Abbreviation explanation not used consequently

minor errors